# Recent Advances on Small-Molecule Antagonists Targeting TLR7

**DOI:** 10.3390/molecules28020634

**Published:** 2023-01-07

**Authors:** Haoyang Zheng, Peiyang Wu, Pierre-Antoine Bonnet

**Affiliations:** 1Faculty of Pharmacy, Montpellier University, 34093 Montpellier, France; 2School of Life Sciences, Shanghai Normal University, Shanghai 200234, China; 3Institut des Biomolécules Max Mousseron IBMM, Ecole Nationale Supérieure de Chimie de Montpellier ENSCM, Montpellier University, Centre National de La Recherche Scientifique CNRS, 34093 Montpellier, France

**Keywords:** TLR7, antagonists, heterocycle, innate immunity

## Abstract

Toll-like receptor 7 (TLR7) is a class of pattern recognition receptors (PRRs) recognizing the pathogen-associated elements and damage and as such is a major player in the innate immune system. TLR7 triggers the release of pro-inflammatory cytokines or type-I interferons (IFN), which is essential for immunoregulation. Increasing reports also highlight that the abnormal activation of endosomal TLR7 is implicated in various immune-related diseases, carcinogenesis as well as the proliferation of human immunodeficiency virus (HIV). Hence, the design and development of potent and selective TLR7 antagonists based on small molecules or oligonucleotides may offer new tools for the prevention and management of such diseases. In this review, we offer an updated overview of the main structural features and therapeutic potential of small-molecule antagonists of TLR7. Various heterocyclic scaffolds targeting TLR7 binding sites are presented: pyrazoloquinoxaline, quinazoline, purine, imidazopyridine, pyridone, benzanilide, pyrazolopyrimidine/pyridine, benzoxazole, indazole, indole, and quinoline. Additionally, their structure-activity relationships (SAR) studies associated with biological activities and protein binding modes are introduced.

## 1. Introduction

Toll-like receptors (TLRs) are a major family of prototypical pattern recognition receptors (PPRs) and type I membranous glycoproteins [1]. TLRs recognize pathogen-associated molecular patterns (PAMPs) and damage-associated molecular patterns (DAMPs). TLRs can initiate antimicrobial host defense responses to restrain pathogenic replication in the innate immune system [2,3]. The engagement of PAMP-PRR interaction allows TLRs to activate downstream signaling molecules of host defense responses [4]. To date, ten TLR subtypes (TLRs 1–10) have been identified in mammals. They are expressed in various innate immune cells, including dendritic cells (DCs), macrophages, and B cells as well as other cell types, such as epithelial cells, endothelial cells, and fibroblasts [5,6]. Notably, TLR7 and TLR9 are only expressed in plasmacytoid DCs rather than myeloid DCs [7]. Several nucleotide-sensing intracellular TLRs, including TLR3, TLR7, TLR8, and TLR9, originally synthesized in the endoplasmic reticulum (ER), are finally transferred to the endosomal compartments [8,9].

TLR7 are selectively activated by guanosine and uridine-containing single-stranded RNA (ssRNA) from viruses, bacteria, endogenous RNA, and oligoribonucleotides in the DCs’ endolysosomes [10]. TLR7 also respond to various chemical ligands, such as small heterocyclic molecules. Such recognition promotes the release of pro-inflammatory cytokines, chemokines, and type-I interferons (IFN), which are involved in the up-regulation of inflammatory reactions [11].

TLR7 is a key determinant of the protective immunity, conversely, its dysregulation is linked to the susceptibility of inflammatory diseases, such as lupus, caused by activation of host-origin nucleic-sensing pathways via TLR7 [12]. Therefore, the rational design of antagonist ligands is a primary focus for the management of autoimmune disorders, cancers, virus infection, and other potential TLR7-associated clinical disorders [13,14].

The development of small-molecule antagonists of TLR7 might start with the identification of chemical scaffolds by high-throughput screening (HTS), or chemical switches, that transform an existing agonist to an antagonist [15]. After the determination of the chemical scaffolds, structure-activity relationship (SAR) studies are then applied to optimize their antagonistic activities and might be followed by the co-crystallization of the inhibitor/TLR7 complexes [16]. Biological studies usually use HEK-Blue or HEK293 cell lines, which are engineered to overexpress TLR7 to indirectly report the nuclear factor-κB (NF-κB) translocation to the cell nucleus. Tests, such as secreted embryonic alkaline phosphatase (SEAP)-driven assay, cell proliferation assay, isothermal titration calorimetry (ITC), and immunoblotting, are also used in in vitro validations. A non-selective TLR7 agonist R848 (Resiquimod) can be commonly used as a positive control [17]. The inhibition of TLR7-induced pro-inflammatory cytokines can be measured through a real-time polymerase chain reaction (RT-PCR) [18].

## 2. TLR7 Main Features

### 2.1. Structural Studies of TLR7

TLR7 is characterized by three distinct domains (Figure 1a) [19]. Firstly, an N-terminal ectodomain (ECD) contains 26 leucine-rich repeats (LRRs). They form a large parallel β-sheet, which lies in the inner part of the ECD allowing a protein–protein interaction between the two monomers. Opposite the β-sheet, α-helices form the convex surface. The different widths between the β-sheet and the α-helices might explain why the ECD structure is curved [20]. The unstructured Z-loop region between LRR14 and LRR15 is particularly important for TLR dimerization. TLR7 requires proteolytic cleavage at the Z-loop for its activation; TLR7 with an uncleaved Z-loop is unable to form the dimer and recognize any microbial RNA [21]. After proteolytic cleavage at the Z-loop region, the N-terminal remains connected with the C-terminal of TLR7 through a disulfide bond between Cys98 (N-ter) and Cys475 (C-ter) [22].

Following ligand stimulation, two ECDs of TLR7 form an m-shape symmetrical homodimer, due to the proximity of the LRR loops. This stage is crucial to trigger a downstream signal transduction [25,26]. Moreover, TLR7 has been considered as a dual receptor. Small-molecule ligands insert into the binding site within the dimerization interface, whereas the binding of oligonucleotides is found at the concave surface, respectively (Figure 1b) [24,27].

Additionally, a transmembrane (TM) domain consisting of a single long transmembrane helix inserts into the lipid endomembrane, due to hydrophobic forces [23]. Finally, the cytoplasmic toll/interleukin-1 receptor (TIR) homology domain, a horseshoe structure, is highly conserved in the TLRs. The TIR domain is taken into consideration for interacting with the other TIR-containing systems to activate a signaling cascade through recruitment of adaptor proteins. In TLR1 and TLR10, the TIR domain has a central parallel five-stranded β-sheet flanked with five α-helices. Inside of the TIR domain, a BB-loop connects a β strand and an α helix, which play a crucial role in the formation of the dimer and activation of downstream signaling. At present, the structure of TIR domain has not been reported yet in TLR7 [28,29].

### 2.2. TLR7 Signaling Pathways

The homodimerization of TLR7 allows to initiate the myeloid differentiation primary response 88 (MyD88)-dependent pathway in the plasmacytoid DCs (Figure 1c) [30]. In the signaling pathway, TLR7 cooperates with protein kinases, transcription factors, and adaptor proteins [31].

Upon ligand binding, TLR7 firstly moves toward the MyD88 adaptor-like protein (MAL) and interacts with MyD88 [32]. Then, MyD88 recruits interleukin-1 receptor-associated kinase (IRAK) family members and forms a large intracellular oligomeric signaling complex: the myddosome. During the formation of the myddosome, IRAK4 activates the autophosphorylation of IRAK1, which is then released to interact with the tumor necrosis factor receptor–associated factor 6 (TRAF6) [33]. TRAF6 functions as an E3 ubiquitin ligase and promotes the non-degradative K63-linked ubiquitination of growth factor-β-activated kinase 1 (TAK1).

Subsequently, the poly-ubiquitinated TAK1 is activated after the formation of a complex with TAK1-binding proteins (TAB1, TAB2, and TAB3) [34]. TAK1 downstream cascades are then divided into two different signaling pathways: NF-κB pathway and mitogen-activated protein kinases (MAPK) pathway [35]. TAK1 phosphorylates IκB kinase β (IKKβ). IKKβ forms a complex with a catalytic subunit IKKα and a regulatory subunit NEMO termed as IKKγ. The IKK complex then phosphorylates NF-κB inhibitory protein IκBα. The inhibitory protein IKK family inactivates and keeps apart the transcription factor NF-κB dimer [36]. Subsequently, the degradation of both IκBα and IκBβ allows the nuclear translocation of the NF-κB, which stimulates the genes encoding IFNs and pro-inflammatory cytokines [37]. Additionally, TAK1 activates the AP-1 transcription factor, which leads to an increased expression of cytokines and IFNβ in the nucleus via the MAPK signaling pathway [1].

## 3. TLR7 Implication in a Variety of Clinical Diseases

### 3.1. Autoimmune Disorders

TLR7 MyD88-dependant signaling pathway drives the production of type 1 IFN in human pDCs and is implicated in the pathogenesis of autoimmune diseases [38]. The abnormal immune system turns its defenses against pathogens upon normal physiological components of the body [39].

Among them, systemic lupus erythematosus (SLE) is a polygenic autoimmune disease characterized by the elevation of two cell types; they are autoreactive age-associated B cells (ABCs) and extrafollicular helper T cells [40]. Additionally, SLE is also associated with the production of antinuclear autoantibodies in multiple organs [41]. Abnormal resistance to the degradation of self-derived RNA activates the TLR7 MyD88 signaling pathway and increases the production of pro-inflammatory cytokines [42]. In SLE pathogenesis, TLR7 can induce the transcription of IFN-stimulated genes (ISGs), which can up-regulate type 1 IFN and activate B cells [43,44]. Several data support the link between TLR7 signaling and B cell activation and production of autoantibodies [45,46]. Similarly, the TLR7^Y264H^ variant resulted in the activation of DCs to release serum lgG in mouse and caused severe lupus in child [39,47]. Up-regulation of TLR7 also increases IFN-β production in pDCs of SLE patients [48]. A recent study also revealed that such a variant increased the affinity of TLR7 for guanosine and cGMP and caused B-cell driven autoimmunity [49]. The TLR7 copy number tightly correlates with disease severity in SLE [50]. In addition, enhanced TLR7 signaling is associated with the differentiation of inflammatory hemophagocytes (iHPCs), which are responsible for anemia and thrombocytopenia in immunity-related diseases [51].

For the management of autoimmune disorders, several studies report the use of synthesized oligonucleotides to act as immunosuppressor of TLR7 [52,53]. Among them, IMO-3100, a TLR7/9 dual antagonist of TLR7 and TLR9 can block the expression of IFN-β, TNF-α, and interleukin 17 (IL-17) and attenuate SLE, rheumatoid arthritis (RA) in a murine model [54]. IMO-3100 significantly improved the expression profile of disease-related MAD-3 genes involved in spindle-assembly checkpoint (SAC) [55]. Recently, IMO-3100 was evaluated in a 4-week phase 2 trial in psoriasis patients (NCT01622348). IMO-9200, a trinary antagonist of TLRs 7/8/9, ameliorates SLE progression in mouse model and shows safety and tolerance in healthy subjects [56]. IMO-8400 is also a dual antagonist of TLRs 7/9 that completed a phase 2 trial in psoriasis pathogenesis (NCT01899729) and dermatomyositis (NCT02612857). The macromolecules will not be described in detail in this review, as its main scope and purpose is to focus on small molecules.

Endosomal TLR7 is also implicated in the pathogenesis of organ-specific type 1 diabetes (T1D), which leads to insufficient insulin secretion and hyperglycemia [57]. T1D occurs as a consequence of the immune destruction of insulin-producing β-islet cells within the pancreas [58,59]. Further to the cytokines release, CD4+ helper T cells promote CD8+ cytotoxic T cells responses, which lead to the β-islet cells blast. Consequently, the loss of β-islet cells makes it difficult for the body to metabolize glucose [60,61]. TLR7 stimulates the up-regulation of a proinflammatory cytokine and type I/II IFNs and accelerates spontaneous onset of autoimmune diabetes [59]. Additionally, TLR7 deficiency suppresses the development of T1D by altering B-cell functions and immunoglobulin production in diabetogenic mouse. Meanwhile, TLR7 deficiency limits the number of CD4+ T cells and reduces the proliferation of antigen-specific CD8+ T cells [58].

Sjogren’s syndrome (SS) is also associated with TLR7 expression [62,63]. SS is a rheumatoid autoimmune disorder characterized by dry eyes and a dry mouth, and often accompanies lupus and rheumatoid arthritis. In this disease, the immune system destroys moisture-secreting glands, such as salivary and lacrimal glands. Patients with SS exhibit increased secretion of inflammatory cytokines in line with TLR7 and TLR9 activation. Such stimulation in peripheral blood B-cells indicates altered TLR signaling [64]. Therefore, the development of antagonists targeting TLR7 might prove beneficial for the treatment of SLE, T1D, and SS.

### 3.2. Immuno-Oncology

TLRs are also involved in the development of various tumors [65]. Either hyperactivation or hypoactivation of TLRs increase the survival and metastasis of a tumor. On the one hand, elevated expression of TLRs signaling induces the production of cytokines and stimulates immune cells, such as DCs, to foster tumor immunotherapy [66,67]. On the other hand, activation of the TLR7 Myd88 signaling pathway induces chronic inflammation, which is an important factor for further putative tumorigenesis and tumor progression [68]. Additionally, TLRs aberrant stimulation could be involved in the early initiation, carcinogenesis, and therapeutic resistance in several types of cancer, such as gastrointestinal malignancies, melanoma, and esophageal cancer [69,70,71]. Dysregulation of TLRs could also enhance immune escape and angiogenesis [72].

Furthermore, TLR7 overexpression is related to high cell proliferation in lung cancer as well as pancreatic cancer [70,72]. Up-regulation of TLR7 decreases the expression of several antitumor molecules implicated in apoptosis. Moreover, increased TLR7 accelerates the proliferation of human CD4+ T helper cells and induces the production of IL-10, IL-2, and IFNγ, and leads to chemoresistance in primary tumors [73,74]. Finally, TLR7 and TLR8 stimulation are associated with immune evasion in line with an increase in the nuclear factor NF-κB and cyclooxygenase-2 (COX-2) expression [75].

### 3.3. Antiviral Immunotherapy and Infection

Endosomal TLRs might promote human immunodeficiency virus type 1 (HIV-1) replication and latency reversal via the stimulation of inflammatory responses [76]. TLR7 overexpression leads to the hypo-responsiveness of CD4+ T cells, and the production of IFN-α in HIV-1 replication [77,78]. TLR7 engagement in CD4+ T cells results in the dephosphorylation of transcription factor NFATc2 and then induces an anergic gene-expression program. In contrast, the anergy of CD4+ T cells could be eliminated by silencing TLR7 [79]. Furthermore, TLR7 and TLR9 were found to be involved in T cell CD95/Fas-mediated apoptosis by inducing Type 1 IFN upon exposure to HIV-1. This enhanced apoptosis has been shown to be inhibited by a phosphonothioate deoxyribose compound acting as a TLR7/9 specific antagonist [80]. Furthermore, an oligonucleotide TLR7/9 antagonist had the potential to abolish the production of virus-induced chemokines, such as interferon gamma-induced protein 10 (IP-10) in HIV-1 viremia [81].

By blocking related cytokines, influenza-related immunopathology could be moderated [82]. The mortality of viral respiratory diseases is often associated with the ‘cytokine storm’ along with an excessive pro-inflammatory cytokine production [83]. TLR7 antagonism demonstrated an adjustive role in the protection of an IFN-1-driven cytokine storm produced by pDCs and monocytes. TLR7 abrogation also reduced the number of lung neutrophils and attenuated inflammation and mortality in influenza in a murine model [84].

TLR7 is also implicated in other infection diseases, such as Pseudomonas aeruginosa pneumonia or Helicobacter pylori infection, and TLR7 or TLR7/8 antagonist might play a positive role in bacterial recognition and treatment [85,86,87].

### 3.4. Others

TLR7 is involved in the pathogenesis of knee osteoarthritis (OA) pain induced by microRNAs (miRNAs) [88]. TLR7 can detect the GU-rich motif of miRNAs; therefore, removal of this TLR recognition motif eliminates OA and blocks the analgesic effect [89]. Clinical data in humans have also suggested a possible involvement of TLR7 in atherosclerotic lesions characterized by the accumulations of lipid, cells and matrix components [90]. The knockout of TLR7 in vivo demonstrated a protective effect toward the atherosclerotic lesions by constraining inflammatory macrophage activation and cytokine production [91].

## 4. Small Molecule Antagonistic Ligands of TLR7

### 4.1. Imiquimod Analogs

Imiquimod **1** was approved by FDA in 1997 and EMA in 1998 (Figure 2) [92]. Imiquimod activates a series of cytokines, such as interferon-α (IFN-α) and IL-6 via TLR7 [93]. At present, imiquimod is commonly used to treat genital warts and superficial basal cell carcinoma (BCC) under the brand name Aldara [94,95].

Recently, our team synthesized three heterocyclic series derived from the chemical scaffold of imiquimod (imidazo [4,5-*c*]quinoline): imidazo [1,2-*a*]pyrazine, imidazo [1,5-*a*]quinoxaline, and pyrazolo [1,5-*a*]quinoxaline [96]. Although the heterocycle core is different from the one from imiquimod, the compounds possess some structural similarities, including an amine group and a hydrophobic alkyl chain. Most interestingly, compounds did not show any agonist property but potent and specific antagonistic activities. Among these heterocyclic series, two pyrazolo [1,5-*a*]quinoxaline derivatives **2** and **3** were identified as lead compounds with IC_50_ = 8.2 μM and IC_50_ = 10 μM for TLR7 inhibition activity while they did not exhibit any activity on TLR8. In comparison to the imidazo [1,2-*a*]pyrazine series, the antagonistic activity of pyrazolo [1,5-*a*]quinoxaline compounds was increased when their alkyl chains reached 4–5 carbon atoms maximum. Meanwhile, the presence of a butyl or an isobutyl chain, such as in compounds **2** and **3**, was shown to be beneficial to obtain the greatest antagonistic potency.

Docking simulation was also performed in the active region of TLR8 (PDB ID: 5WYZ), due to the high sequence similarity and similar antagonistic site between TLR7 and TLR8. Thereafter, a box covering the likely TLR7 binding antagonist sites was applied, which allowed to determine various chemical interactions located in the interface of TLR7 dimer. The tricyclic scaffold of the pyrazoloquinoxaline forms a face-to-face π-π stacking with Phe1302, followed by additional van der Waals forces with Tyr242. In addition, the alkyl chain was engaged into the hydrophobic pocket of TLR7. In particular, Thr384 of TLR7 was indicated to explain the selectivity in comparison with Ile403 of TLR8 containing a bulky side chain, regardless of the high degree of similarity between TLR7 and TLR8. In in vitro tests, compounds **2** and **3** inhibited the expression of TNF-α and interleukin 6 (IL-6) in HEK-blue-hTLR7/8 cells. Additionally, compounds **2** and **3** selectively inhibited NF-κB translocation in TLR7 MyD88 signaling pathways.

A recent study showed also that TLR7 can adopt both imidazo [4,5*-c*]quinoline-based agonists and antagonists. A C2-alkyl substitution on the tricyclic core can interact with a hydrophobic area of the receptor at the dimer interface and appeared to be a determinant for its activity [97].

### 4.2. Quinazoline-Based Ligands

The quinazoline derivative CPG-52364 **4** is a potent trinary TLR7/8/9 antagonist with a ratio for TLR7/9 antagonism of 0.8 (Figure 2) [54]. CPG-52364 was tested for tolerability in chronic autoimmune diseases, including SLE, but unfortunately failed in phase 1 clinical trials (NCT00547014). The reason of discontinuation remains unpublished.

Recently, some new TLR7 antagonists with the same quinazoline scaffold were synthesized [98]. The development of diverse TLR7 antagonists was mainly accomplished through random screening. To correlate TLR7 antagonistic activity with the structural features in different chemotypes, the authors derived a hypothetical binding model based on molecular docking analysis along with molecular dynamics (MD) simulation studies by using the homologue protein hTLR8 (PDB ID: 3W3J). Binding studies exhibited different pockets, grooves, and a central cavity where the ligand could interact with specific residues of its receptor through hydrophobic and hydrogen bonds. Molecular docking analyses determined the presence of three hydrophobic pockets and two small grooves in the binding site of TLR7 antagonist **4**.

Such studies paved the way for the rational design of several chemotypes. Based on the structural insight gained, TLR7 antagonists with the quinazoline core were designed to better understand the engagement mode of the molecules within the diverse protein pockets. The biological evaluation of the synthesized molecules was performed in TLR7-reporter HEK293 cells as well as in pDCs. The study provided a rational design approach thus facilitating further development of novel small molecule hTLR7 antagonists based on different chemical scaffolds. Among them, the best active molecule **5** showed TLR7 inhibition with an IC_50_ of 1.03 μM, and the suppression of IFN-α induction in response to TLR7 activation in pDCs with an IC_50_ of 1.42 μM. Separate substitution of quinazoline at C2 or C7 position with a flexible linker and the presence of other small hydrophobic groups were shown to be beneficial for achieving potent TLR7 antagonism. Meanwhile, the cooperation of C2, C4, and C7 substitution could lead to an increase in antagonistic activity of TLR7.

With the most potent antagonist of quinazoline derivatives, molecular dynamics simulation studies show highly stable interactions, which lead to a steady-state fold of TLR7. This study highlights the crucial role of hydrophobic interactions around the central cavity of the antagonist site with important residues, such as Tyr356, Thr525, and Gln354 in the binding mode of the most potent antagonist derivatives.

### 4.3. Purine-Based Ligands

TLR7 can also respond to guanosine and to small molecules containing a purine core. Such data were of great value for rational drug design [24,99]. An adenine scaffold was previously reported in a series of TLR7 agonists [100]. Hydroxyl groups at C8 and C2 were demonstrated to be important to boost IFN activity [101]. Meanwhile, an amino group at C6 and the introduction of an alkylamino group on the N9 position increased aqueous solubility [102]. Such chemical modification on the adenine core were determined to be useful for antagonism, as well as drug delivery and absorption.

In 2020, Tojo S. et al. developed, for potential SLE applications, a series of TLR7 antagonists based on the 8-oxoadenine core using a chemical switch approach [103]. Based on SAR studies, the authors emphasized that the C8 position of adenine could be substituted from an 8-oxo to an 8-pyridyl group, which allowed the conversion of an agonist into an antagonist. Additionally, the C2 side chain and 8-pyridyl moiety were further optimized, and an amine was introduced on the para-position of the 9-phenyl group in order to increase solubility. As a result, a potent TLR7 antagonist **6** based on a 6-methyl derivative was successfully synthesized with high inhibitory activity (IC_50_ = 15 nM), selectivity and good solubility (>0.15 mg/mL at pH 7.4) (Figure 3).

Crystallographic studies showed that the most active TLR antagonist **6** interacted at the dimerization interface of the TLR7 protein and induced an open form conformation of the dimer. Such an open form, which shows large differences at the interface compared to the closed form, appears to prevent TLR activation. The 6-methyl adenine and 8-fluoropyridine rings of the most active antagonistic compound are capable to form π-π stacking with Phe408, Phe506, Phe507, Phe349, Phe351, Leu353, and Val381. A hydrogen bond was observed with Gln354, and additional interactions were depicted with Gln323, Phe349, Tyr264, Asn265, Phe351, Ser530, and Tyr579. In vitro results showed that compound **6** totally blocked IFN-α production induced by R848, which suggested compound **6** was specific in inhibiting TLR7. Furthermore, in vivo studies showed compound **6** significantly decreased proteinuria and prevented death in SLE mice.

In 2020, Mukherjee A. et al. reported a very simple modification at the 2-position of 9-alkylpiperazinyl guanosine derivatives induced to a chemical switch from a TLR7 agonist **7** to a clinically relevant antagonist [15].

The deletion of a butoxy moiety afforded compounds with antagonist activity, leading to a potent TLR7 inhibitor **8** after having further increased lipophilic interactions by modifying the amino group on the 6-position. Compound **8** exhibited an IC_50_ of 4.7 μM on TLR7 and induced inhibition of IL-6 transcription and proinflammatory cytokines, such as TNF-α. It also showed significant potency on an in vivo rodent model of psoriasis. In a molecular docking study, the team used the homology model of TLR7 (PDB ID: 5WYZ) and showed that the purine core formed π-π stacking interactions between two hydrophobic residues, Phe329 and Tyr1374. Meanwhile, ethylpiperazine at C6 formed an important hydrogen bonding between the terminal nitrogen and Ser452. Additionally, the alkyl linker on N9 interacted with Leu331 and Phe329 in a hydrophobic cavity.

In a further study, the same team developed an orally bioavailable dual TLR7/9 antagonist **9**, with IC_50_ values of 0.08 and 2.66 μM against TLR9 and TLR7, respectively [104]. They highlighted several suitable substitutions at C2, C6, and N9 positions that could exhibit considerable efficacy in TLR7 and TLR9 antagonism. Compared to compound **8**, compound **9** bears a 4-methoxybenzyl group in the C2 position and a pyrrolidine group, instead of pyrazine, at N9. After in vivo evaluation, compound **9** was suggested to be available as a therapeutic candidate in autoimmune disorders after being shown to have therapeutic effects on psoriasis in mice.

### 4.4. Imidazopyridine-Based Ligands

Recently, Das N. et al. developed a potent dual TLR7/9 antagonist based on a 2-phenylimidazopyridine heterocyclic core with a bridgehead nitrogen, which exhibited IC_50_ values of 0.47 and 0.04 μM against TLR7 and TLR9, respectively [105]. They initially intended to explore minimal pharmacophoric features around the basic core **10** (Figure 4). The presence of two methoxy groups at C2 and C4 positions **11** on the phenyl group and a NH_2_ function at C7 position of the heterocyclic platform showed interesting TLR7 antagonistic activity with an IC_50_ of 2.55 μM. Among these modifications, the NH_2_ group at the C7 position acts as a hydrogen bond acceptor, which stabilizes the pharmacophore in the binding site. 

Concurrently, the imidazopyridine core forms π-π interactions with the central hydrophobic cavity. The oxygen atom of the methoxy group at the C4 position forms a hydrogen bond contributing to the stabilization of the ligand-receptor combination. Di-methoxy substitution on the 2-phenyl ring leads to a compound **12**, which exhibited TLR9 and TLR7 inhibition with IC_50_ values of 40 and 466 nM, respectively. After replacement of the NH2 group by an alkyl-substituted piperazine and the introduction of a pyrrolidine cycle at position 3 of the imidazopyridine core, Das N. et al. obtained a lead compound **13**, which exhibited TLR9 and TLR7 inhibition with IC_50_ values of 15 and 258 nM, respectively.

### 4.5. Pyridone-Based Ligands

Knoepfel T. et al. developed a selective dual TLR7/8 antagonist further with high-throughput screens (HTS) using TR-FRET (time-resolved fluorescence resonance energy transfer) assays with a biotinylated TLR8 protein and Eu-labeled streptavidin [16]. Consequently, they first identified a hit compound **14** with a quinazoline core (Figure 5) forming face-to-edge stacking located in a hydrophobic pocket and a pyridin-2-one group for hydrogen interactions. The hit-to-lead optimization process showed that 8-methyl substitution **15** increased the cellular potency and affinity toward TLR8 but was detrimental to TLR7 antagonistic activity. Therefore, even though TLR7 and TLR8 have a high degree of structural similarity, two main differences were noted on two non-conserved amino acids between TLR7 and TLR8. In comparison to TLR8 (PDB ID: 6TY5), alanine 518 is a serine (530) in one of the TLR7 monomers, and glutamic acid 427 is a valine (430) in TLR7 in the other one. Considering such differences, they changed the quinazoline core by an indazole to avoid the interaction with Glu427 of TLR8 and a steric clash with Ser530 of TLR7. The next step was to separate TLR7/8 balanced activity from TLRs 4/9. A 6-methyl substitution **16** on the pyridone exhibited desired TLR7/8 potency without any TLR4/9 profile and showed the C6 position of the pyridone was crucial for TLR4/9 activity. For the final step of their lead optimization, they focused on the binding to TLR8, since TLR7 activity was still maintained in order to obtain a dual TLR7/8 antagonist **17**, with very low IC_50s_ = 0.62 and 1.5 nM for TLR7 and TLR8, respectively.

### 4.6. Benzanilide-Based Ligands

Padilla-Salinas R. et al. performed a first high-throughput screening (HTS) targeting TLR8, which allowed them to identify two first hits, a non-benzanilide TLR8-specific inhibitor **18a** and a 2,6-dihalogenobenzanilide derivative **18b** with TLR7/8 dual inhibitory activity (Figure 6) [106]. A series of SAR studies mainly performed on the benzanilide derivative showed the importance of the presence of three closed linked phenyl rings, with a separation by an amide group for two of them, one of the phenyl rings being substituted by two trifluoromethyl groups. Furthermore, an optimal distance had to be maintained between the different aromatic rings. Benzanilide ring modulation was intolerant for TLR7. Replacement of a trifluoromethyl group by a trifluoromethyl pyridine led to the TLRs 7/8 antagonist CU-72 **18c**, which showed micromolar inhibitory activities on TLR7 (IC_50_ = 5.1 μM) and TLR8 (IC_50_ = 2.87 μM). In vitro studies suggested CU-72 could act for the inhibition of ssRNA-sensing pathways at low concentrations. Unfortunately, the compound shows low solubility and remains to be further optimized for extended biological evaluation.

### 4.7. Pyrazolopyrimidine/Pyridine-Based Ligands

Alper P.B. et al., using a large screening study performed on a databank of two million compounds, further identified a first hit pyrazolopyrimidine compound **19** with a piperazine moiety (Figure 6) as a good inhibitor of the TLR7 agonist imiquimod on IL6 production [107]. TLR7 antagonistic activity was then shown to be significantly increased by replacement of the piperazine by a piperidine ring **20** and then a bicyclo [2,2,2]octane. The next step was to focus on chemical modulations to improve the oral exposure and solubility. Replacement of the pyrazolopyrimidine scaffold by a pyrazolopyridine one and a change in the attachment position to the heterocycle platform afforded the lead compound **21** (IC_50_ < 1 nM).

### 4.8. Benzoxazole-Based Ligands

Lamphie, M. et al. studied the activity and mechanism of the action of two potent and selective benzo[d]oxazole TLR7/9 antagonists, AT791 **22** and E6446 **23** (Figure 7) [108]. The two compounds were initially developed to block TLR9 DNA-induced stimulation, but they were also shown to inhibit TLR7 stimulation. Such inhibition does not appear to be linked to a direct binding to TLRs but by their accumulation as lipophilic weak bases in endosomes, as, for example, antimalarial drugs and, more interestingly, their high and selective affinity for nucleic acids.

### 4.9. Indazole-Based Ligands

Betschart C. et al. reported a potent and selective dual TLR7/8 antagonist **24** with IC_50_ values of 0.39 and 0.08 nM on TLR7 and TLR8, respectively (Figure 7) [109]. Their study started from a fragment-based screening, and hits were optimized following SAR and co-crystallography with TLR8 (PDB ID: 7R53 for compound **24**). Such latter data showed that compound **24** stabilizes the inactive conformation of TLR8, and indazole formed two key hydrogen bonds with Gly351 in one monomer and Ser516 in the other one. The cyclopropyl ring of antagonist **24** occupies a nearby hydrophobic pocket. In in vitro and in vivo studies, compound **24** showed inhibition of TLR7-dependent IFNα release.

### 4.10. Indole-Based Ligands

Afimetoran **25** (BMS-986256) is an oral potent inhibitor of TLR 7/8 in clinical trials (Figure 8). It is an indole-based compound with two heterocyclic substituents, a triazolopyridine- and a piperidine-based structures. Afimetoran had the ability to prevent lupus symptoms in mouse models and reverse organ damage, and it showed promising PK data as well as safety profiles in a phase 1 trial (NCT04493541). A recent phase 2 study evaluates its efficacy in patients with systemic lupus erythematosus (NCT04895696).

In 2020, a BMS group of medicinal chemists reported the discovery process of a series of indole-based TLR antagonists, with the exclusion of BMS-986256 [110]. Their SAR study concentrates on tri-substituted indole derivatives with, for the most active compounds, a dimethoxy phenyl substitution on position 2, a short alkyl group on position 3, and the piperidinyl heterocycle grafted on position 5. The most active compound **26** was determined to be a very potent and dual TLR7/8 inhibitor with IC_50_ values of 10 and 17 nM, respectively. It was also shown to have a much lower effect on TLR9 activity.

Recently, in continuum of this first SAR study, the same group developed a 2-pyridinyl-indole series that allows them to afford a compound **27**, BMS-905, another dual inhibitor of TLR7/8 [111]. In this study, they demonstrated that the aryl substitution on position 2 of the indole was important for ligand–protein interactions. They firstly determined the engagement of the heteroatom with various pyridinyl isomers. The 2′,6′-dimethyl groups on the pyridinyl group induced the greatest potency in both TLR7 and TLR8. The substitution of the C3 position of the indole by an isopropyl was shown to be essential for selectivity toward TLR9. With these changes, BMS-905 was discovered as a second potent TLR7/8 dual antagonist lead, with IC_50_ values of 0.7 and 3.2 nM for TLR7 and TLR8, respectively. It is interesting to note that two of the three main structural features of compound **26** and BMS-905 can also be found in BMS-986256 with an important change on position 2 by replacement of the phenyl or pyridinyl group by the triazolopyridinyl heterocycle.

### 4.11. Quinoline-Based Ligands

Enpatoran **28** (M5049) is a 5-piperidylquinoline derivative developed as a potent and selective dual TLR7/8 antagonist with IC_50_ values of 11.1 and 24.1 nM for TLR7 and TLR8, respectively (Figure 8). M5049 appears to bind the open form of the TLR dimer and stabilizes the inactive state of the receptor, which prevents its activation [112].

In in vitro and in vivo tests, enpatoran shows a therapeutic effect in SLE, rare cutaneous lupus erythematosus (CLE), and other autoimmune diseases by suppressing the pathologic activity of ribonucleic acid–containing immune complexes. In comparation to placebo, M5049 showed promising drug-like properties with good tolerance and promising PK without significant dose-limiting adverse events in healthy volunteers. Further studies focus on the evaluation of its potential efficacy in patients with autoimmune diseases, such as lupus, with high TLR7 and TLR8 expression [113].

Additionally, TLR7 is also involved in coronavirus disease 2019 (COVID-19), due to the participation of TLR7 in the invasion and infection of the virus [114,115]. Therefore, enpatoran also underwent a phase 2 trial in order to evaluate its safety and efficacy in COVID-19 pneumonia patients (NCT04448756). High exposure (100 mg) to enpatoran reduced the inflammatory response that leads to the cytokine storm in some cases, and was recommended to reduce acute hyperinflammation caused by COVID-19 [116].

## 5. Conclusions

TLR7 is crucial for recognizing xenobiotics containing PAMPs in the innate immunity system. The TLR7 overexpression, caused by the presence and abnormal degradation of self-RNA, activates MyD88-dependent signaling pathway and guides the host defensive responses. Hence, TLR7 is an attractive target for drug design studies and development. In this review, we provide general information for potential therapeutic treatments, including autoimmune disorders, cancer, and antiviral immunotherapy. Various heterocyclic aromatic scaffolds of small molecule TLR7 antagonists are described, many of them bearing a five-membered cycle. The obtention of the lead compounds include SAR optimization, and some of them are X-ray co-crystallography data. For small molecule antagonists, most of the studies state that TLR7 antagonists occupy the same binding site as agonists and change the closed-form conformation of TLR7 to an opened-form conformation. Several small molecule TLR7 antagonists have undergone clinical trial; the first results appear to assess their safety of use in patients. Afimetoran, an indole-based TLR7 antagonist, and Enpatoran, a quinoline derivative, are under phase II clinical investigation in patients with SLE or other immune related diseases.

## Figures and Tables

**Figure 1 molecules-28-00634-f001:**
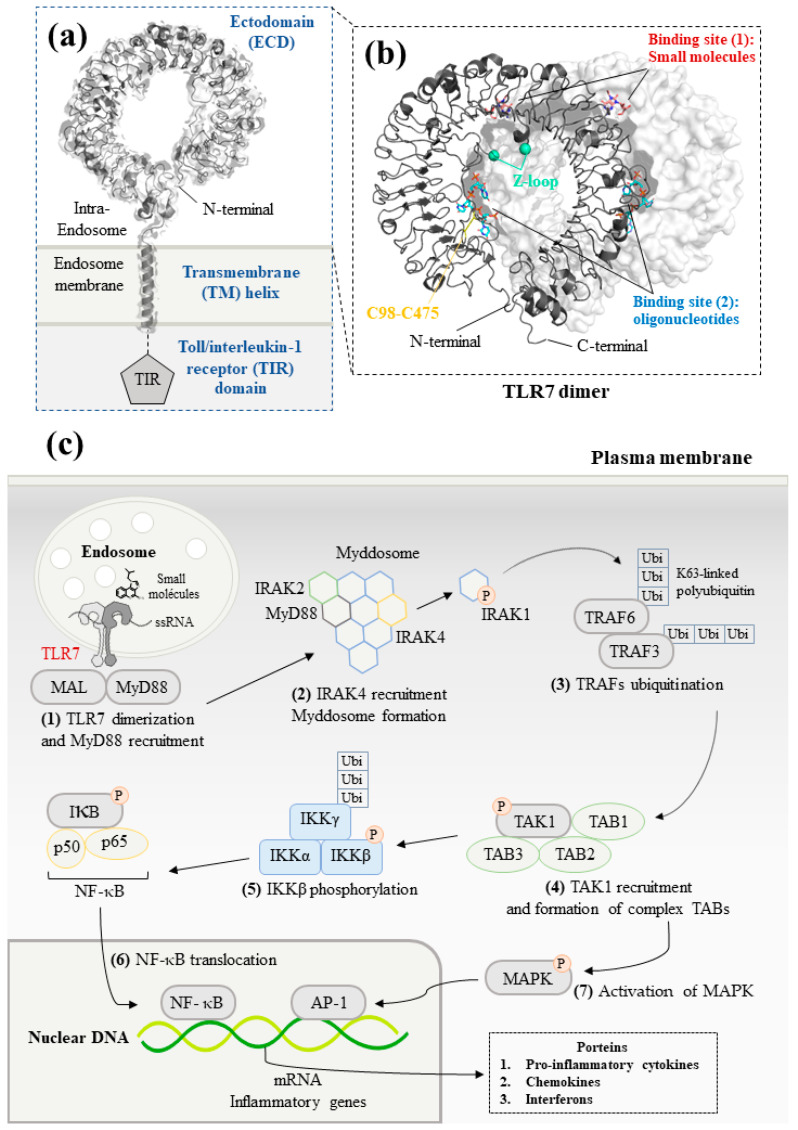
Structure and signaling pathways of the toll-like receptor 7 (TLR7): (**a**) Schematic representation of TLR7. The N-terminal ectodomain (ECD) locates in the endosome. The transmembrane helix consists of an α-helix across the endosomal membrane. The details of the structure of toll/interleukin-1 receptor (TIR) domain are not illustrated due to the lack of data for TLR7. Data are from reference [23] (PDB ID: 7CYN). (**b**) Front view of the ECD dimer. The cleaved Z-loop and a disulfide bridge between Cys98 and Cys475 are shown in cyan and yellow, respectively. The binding site of small molecules and synthetized oligonucleotides are illustrated in red and blue, respectively. Data are from reference [24] (PDB ID: 5GMG). (**c**) TLR7-mediated MyD88-dependent signaling pathways. Firstly, the recognition between small-molecule ligands or ssRNA and TLR7 allows to initiate downstream signaling. MyD88 then forms the myddosome with IRAK4, which phosphorylates and releases IRAK1. After that, TRAF6 is auto-ubiquitinated and activates TAK1. TAK1 forms a complex with TAB1/2/3. Finally, translocation of NF-κB and activation of MAPK signaling pathway generates innate immune responses that lead to the production of pro-inflammatory cytokines and IFNs.

**Figure 2 molecules-28-00634-f002:**
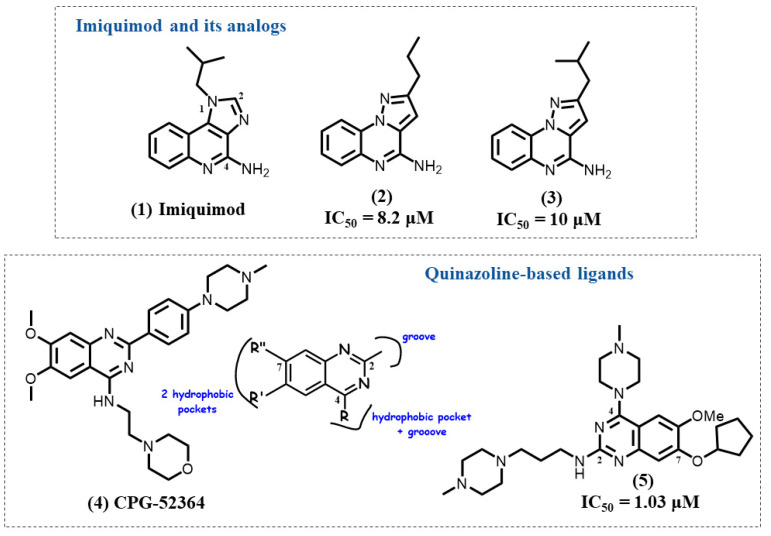
Imiquimod analogs and quinazoline derivatives.

**Figure 3 molecules-28-00634-f003:**
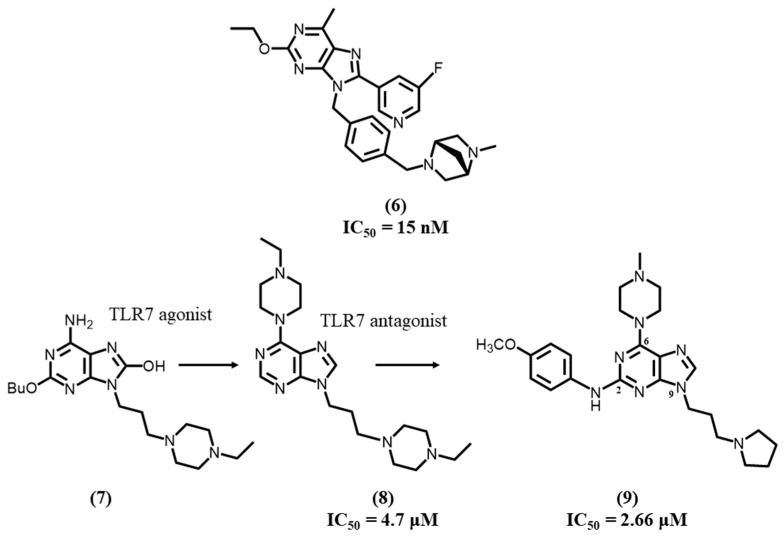
Structure-activities relation (SAR) studies and chemical structures for purine derivatives.

**Figure 4 molecules-28-00634-f004:**
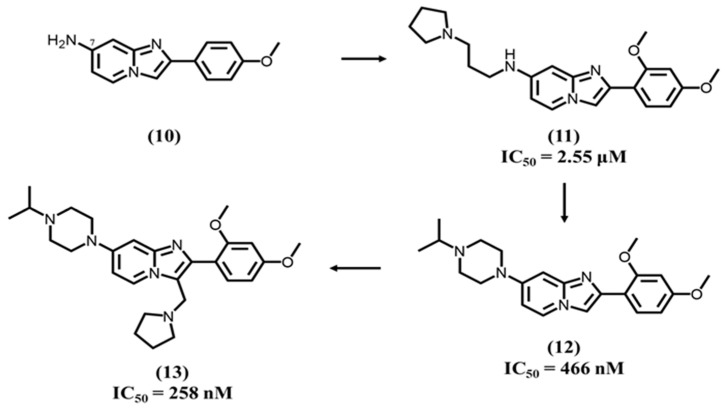
SAR studies on imidazopyridine derivatives.

**Figure 5 molecules-28-00634-f005:**
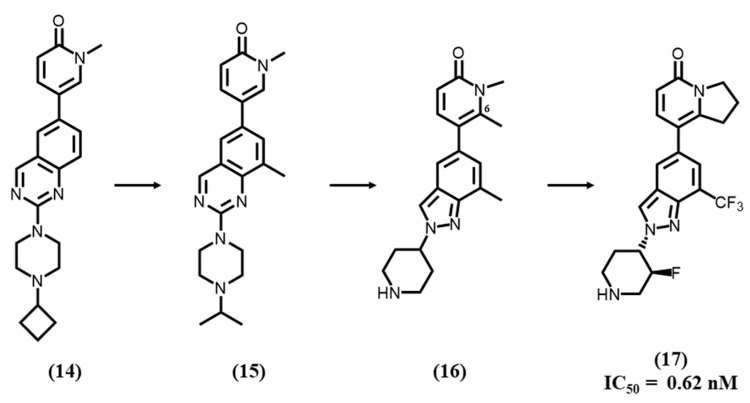
SAR studies with pyridone derivatives.

**Figure 6 molecules-28-00634-f006:**
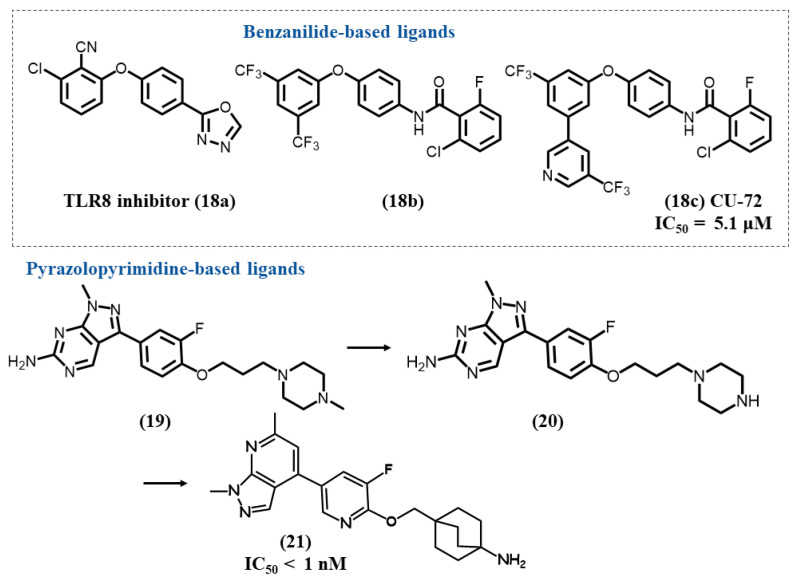
Structures of benzanilide-based ligands and SAR studies of pyrazolopyrimidine/pyridine derivatives.

**Figure 7 molecules-28-00634-f007:**
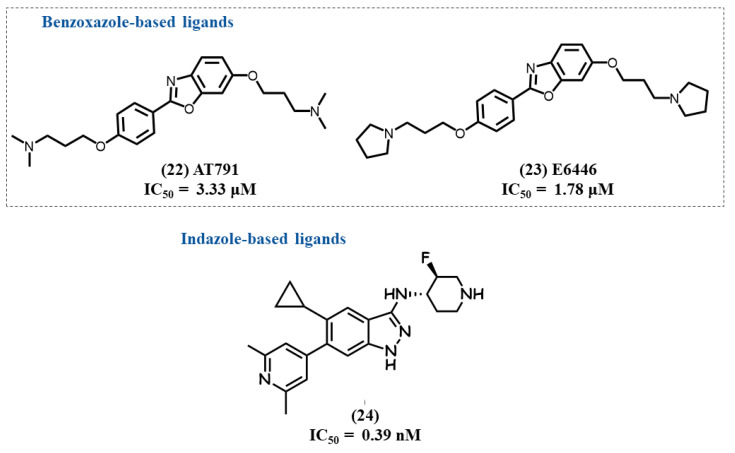
Benzoxazole and indazole derivatives.

**Figure 8 molecules-28-00634-f008:**
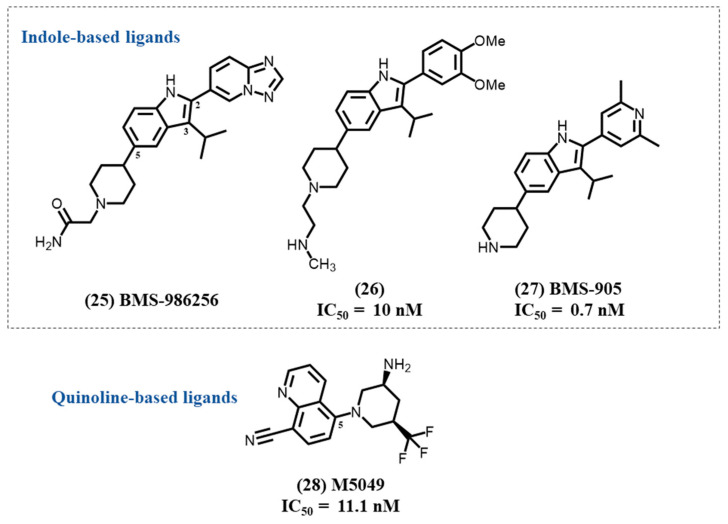
Indole and quinoline TLR7 antagonist derivatives.

## Data Availability

Not applicable.

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
