# Peer review of "Recent Advances on Small-Molecule Antagonists Targeting TLR7"

_molecules, 2023, doi:10.3390/molecules28020634_

Round 1

Author Response

We thank very much the editor and the reviewers for their comments and constructive remarks. All of them have been taken into consideration in the revised version of the manuscript.

Reviewer 1:

Thank you very much for the comment. Implication of TLR in infection diseases has been added in the revised version, end of section 3.3, and suggested references added.

Reviewer 2 Report

Bonnet et al reports recent advances on small-molecule antagonists targeting TLR7. In early part, structural information of TLR7 and biological function and relationship between TLR7 and diseases. After that, small molecule antagonists are summarized.

To understand the state-of-art of the TLR7 research, this manuscript is appropriate and helpful.  However, it is difficult for me to understand the contents which explain only text without figures. For example, in section 4.2, docking analysis of second paragraph is difficult to understand, and to discuss the effect of substituents of compound 5, information of the molecules for comparison should be added in figure. In section 4.6, if the structures of two first hits compounds and some reference compounds of SAR are added, it is very helpful to understand the contents.

Other points;

1. I would like to know the reason that CPG-52364 (4) failed in clinical trials.

2. The values of IC50 each molecules should be added to figures.

3. If the numbers to indicate the position of substituents in compounds are added, it is helpful to understand the contents. For example, position number of adenine in Figure 3. 

Author Response

We thank very much the editor and the reviewers for their comments and constructive remarks. All of them have been taken into consideration in the revised version of the manuscript.

Reviewer 2 :

A schematic representation of the hydrophobic pockets and grooves has been added in Figure 2 in order to better explain and understand the main results from the SAR and MD studies reported in the second paragraph of section 4.2 and explain the effects of substituents of compound 5.

In section 4.6, the structures of the two first hits compounds molecules 18a and 18b have been added for comparison in Figure 6.

The reasons for discontinuation of the clinical trials for CPG-52364 have not been made public and have not been found in the literature. We have added such information at the end of the first paragraph of section 4.2.

The values of the IC50 of each molecule have been added to Figures

The numbers of the most relevant positions have been added for information in several structures of Figures 2, 3, 4, 5 and 8.

Reviewer 3 Report

This manuscript reviews the recent developments in the search for small-molecule antagonists of the TLR7 receptor. TLR7 is relevant to many health disorders and the search for novel ligands for this receptor is of considerable interest. The review is written in good style and sufficient detail. It provides valuable information about the structure and function of TLR7, before proceeding with the various types of small molecule antagonistic ligands. A total of 11 different structural types of antagonists are reviewed, with many interesting examples and SAR studies. Own contributions of the authors are also included.

I only noticed a small technical error: Line 501 “Note applicable” should probably read “Not applicable”

In view of the above, I recommend publication of the manuscript in its present form.

Author Response

We thank very much the editor and the three reviewers for their comments and constructive remarks. All of them have been taken into consideration in the revised version of the manuscript.

Reviewer 3 :

As suggested by the reviewer, english language has been supervised and several improvements in writing made in the revised version of the document.

The error in line 501 has been corrected.

Round 2

Reviewer 1 Report

In the revised form of the article the authors accepted the reviewer's suggestions. The paper can now be accepted.